# Extreme Drought around Tomsk, Russia in Summer 2012 in Comparison with Other Regions in Western Siberia

**Hiroshi Matsuyama** [1,*] , **Takanori Watanabe** [1] **and Valerii Zemtsov** [2]

1   Department of Geography, Tokyo Metropolitan University, Hachioji 192-0397, Japan
2   Department of Hydrology, National Research Tomsk State University, 634050 Tomsk, Russia
*   Correspondence: matuyama@tmu.ac.jp; Tel.: +81-42-677-2603

**Abstract:** The objective of this study is to clarify the regional difference in hydrometeorological parameters in Western Siberia (WS), an area which suffered from severe drought in the summer of 2012. The drought was especially apparent in middle WS. Regional differences in the hydrometeorological parameters have not been fully investigated so far; therefore, we investigated them based on the temporal variation in the hydrometeorological data. All of WS experienced an extremely hot summer in 2012, particularly in June and July. In middle WS, the snow water equivalent in March 2012 was the third lowest recorded from 1985 to 2019. The runoff during April–September 2012 was smaller than the long-term mean. Precipitation during April–August 2012 was also continuously lower. All this resulted in a severe drought in the summer. In particular, precipitation in July 2012 in middle WS was among the lowest recorded for the period of 1966–2019. These characteristics were unique to middle WS in July 2012. North and south WS did not suffer from a severe drought in 2012 because substantial precipitation was observed in summer. The findings of this study will contribute to the prediction of future hydrometeorological events, as extreme phenomena are more likely to occur in accordance with the progress of global warming.

**Keywords:** severe drought; Tomsk; Western Siberia; snow water equivalent; runoff; temperature; precipitation

## 1. Introduction

### 1.1. Severe Drought around Tomsk in Summer 2012

Current changes in the Siberian climate are manifested not only in the increase in near-ground air temperature, but also in the changes in precipitation and increase in frequency/severity of abnormal hydrometeorological events [1]. Nowadays, they sometimes can last several weeks or even months, owing to the formation of stable synoptic structures blocking the westerlies in the atmosphere [2]. Such extreme weather events certainly affect crop productivity as well [3–5]. As an example, this study considers the 2012 extremely long low-flow period in Western Siberia (WS, Figure 1).

A dry autumn in 2011 was followed by lower snowfall in winter and a dry summer with atmospheric precipitation less than 60% of the long-term mean (Figure 2). The extremely low seasonal spring flood in the Tom River reached extreme summer lows in August 2012 [6]. Owing to insufficient filling of the Novosibirsk reservoir on the Ob River, the hydro-energy sector suffered severe damage. In addition, there was difficulty in navigating the rivers of Ob, Tom, and their tributaries [7]. The upper Irtysh cascade of water reservoirs in the southern part of the Ob River basin could not release environmental flows from spring to summer in 2012 [8], and the lakes in the permafrost zone of WS almost dried up [9].

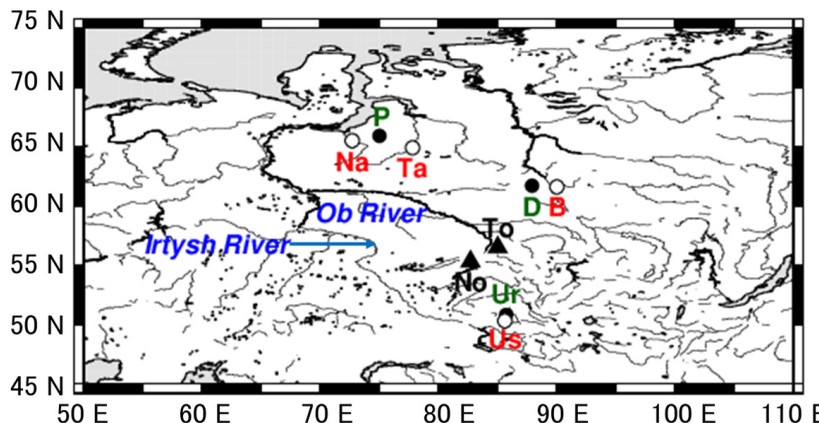

**Figure 1.** Study area. Tomsk and Novosibirsk are represented by black triangles whereas black circles are the experimental basins listed in Table 1. White circles are meteorological stations listed in Table 2. B: Bor, D: Dubches at Sandakches, Na: Nadym, No: Novosibirsk, P: Pravaya Khetta at Pangody, Ta: Tarko-Sale, To: Tomsk, Ur: Ursul at Onguday, Us: Ust-Koksa. Tomsk is located on the right bank of Tom River, a tributary of Ob River.

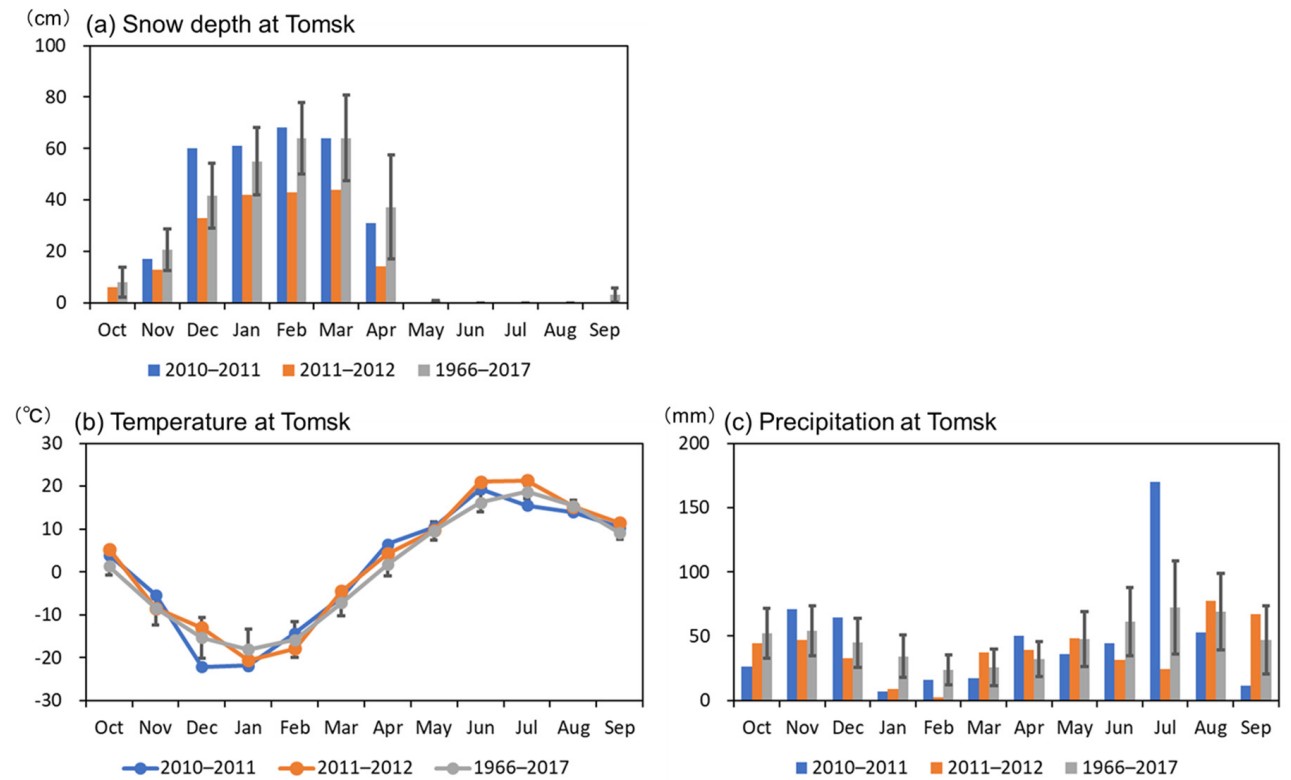

**Figure 2.** Seasonal change in hydrometeorological elements at Tomsk from October to September [10]. The bars indicate the standard deviation of the long-term mean for the respective gray legends. The data for 2010–2011 and 2011–2012 are also displayed. (**a**) Snow depth, (**b**) temperature, and (**c**) precipitation. The details of the hydrometeorological data are listed in Table 1.

In Tomsk, the temperature was approximately +4–5 °C in June 2012 and +2 °C in July 2012 compared to the long-term mean from 1961 to 1990 [11]. The temperature anomaly was more relevant and wider in June than in July, with the temperature in the former reaching as high as +10 °C in Northwestern Siberia [11]. Additionally, precipitation in June and July 2012 was lower in WS. At Tomsk, a precipitation of 33.0 mm and 24.4 mm was observed in June and July 2012 (Figure 2c), respectively, which were 55% and 33%

of the long-term mean, respectively [11]. A summary of information from the Japan Meteorological Agency [12] revealed that in the winter of 2011 to 2012, the westerlies meandered from north to south, and warm air entered WS, where the anticyclone prevailed. WS was covered by a ridge and had little precipitation throughout the cold season. In the summer of 2012, temperatures in the mid- and high-latitude troposphere of the Northern Hemisphere exhibited markedly hot anomalies. In WS, precipitation rarely fell in June and July 2012, although this atmospheric drought was not mentioned in [12,13].

### 1.2. Literature Review on the Severe Drought in WS in Summer 2012

Except for the previous studies mentioned in Section 1.1, some studies have investigated the severe drought in WS in 2012. Abnormally hot and dry weather prevailed from 30 May to 2 August 2012 in the Tomsk region—the highest in the last 60 years [14]. These abnormal conditions damaged the agricultural areas of the region. The characteristics of atmospheric drought were investigated in Southern Siberia from 1979 to 2017 [15]; there has been little significant change in hydrothermal conditions over the past 40 years. However, the duration of dry periods during the growing season has increased. Additionally, the frequency of extreme events—both droughts and downpours—has increased in recent years.

The water quality in thermokarst ponds and lakes during the summer of 2012 was investigated from a hydrological perspective [9]. The results were compared with those of the normal summer in 2010, although Western Russia suffered from heat waves in 2010 [2]. Lake water warming at high latitudes increases the methane emission capacity from thaw lake surfaces and decreases the molecular size of trace elements bound to colloids [9]. Additionally, relatively conservative responses to carbon dioxide, dissolved organic carbon, and trace element concentrations will occur. Another study [16] compared hydrological characteristics, including for 2012, among regions in different climatic zones in Siberia, but not among regions during the same year. The authors [16] indicated that the drought in 2012 changed the runoff characteristics of a few watersheds.

### 1.3. Objective of This Study, and the Organization of This Article

All of the above-mentioned studies were limited to case studies in certain places around Tomsk in 2012. A comparative study on a broader spatial scale is required to highlight the characteristics of severe droughts around Tomsk. The general idea is that the most dangerous extreme hydrologic events might—or even, should be—considered in a wider temporal and spatial context, such as a chain of consecutive events caused by the change in the atmospheric circulation pattern (atmospheric drought—soil drought—hydrologic drought), as reviewed by [17].

Considering this problem, the authors have conducted a Japan–Russia cooperative study entitled, "Comparative analysis of the impact of increasing extreme hydrometeorological events on the carbon and water cycles of the Arctic and alpine landscapes in the context of sustainable development of the northern and mountainous regions" from 2019 to 2022, with financial support from JST/SICORP (Japan Science Technology/Strategic International Collaborative Research Program). V.Z. was supported by a Russian Federal Targeted Program Grant (RFMEFI61419X0002). Based on the results of the cooperative study, we compared the hydrometeorological characteristics of the three different regions in WS in 2012 using long-term data. Thereafter, we highlighted the peculiar characteristics of severe drought around Tomsk in the summer of 2012.

The rest of this article is organized as follows. Section 2 explains the data and methods adopted in this study. Section 3 simultaneously includes the results and discussion which is suitable in the case of this study. The unique characteristics of middle WS in comparison with the other areas of WS are well documented. Section 4 concludes this study and shows future prospects. Again, the objective of this study is to clarify the regional difference in hydrometeorological characteristics in WS in the extreme drought year of 2012. The findings of this study will certainly contribute to predict future hydrometeorological conditions



because extreme phenomena are likely to occur in accordance with the progress of the global warming.

## 2. Data and Methods

First, we investigated the spatiotemporal variation in the atmospheric field during April–September 2012 to confirm the severe drought in middle WS. The variables included precipitation, surface air temperature, and geopotential height at 500 hPa, with a temporal resolution of one month. For precipitation, we used CMAP (Climate Prediction Center Merged Analysis of Precipitation) [18], which is based on gauge observations, satellite estimates, and numerical model outputs. Regarding air temperature and geopotential height, we used the NCEP/NCAR (National Center for Environmental Prediction/National Center for Atmospheric Research) reanalysis [19]. To obtain these data, we utilized the website of NOAA/PSL (National Oceanic and Atmospheric Administration/Physical Science Laboratory) [20]. We drew monthly anomalies from 1961 to 1990 to match the figure in [10].

Next, we investigated the hydrometeorological characteristics of three experimental basins in WS: Pravaya Khetta at Pangody in the north; Dubches at Sandakches in the middle; and Ursul at Onguday in the south (Figure 1). Hydrometeorological data listed in Table 1 were used. The temperature and precipitation in these basins were derived from [10], whereas hydrological data were derived from [21]. As for the snow water equivalent, observations were conducted approximately every 10 days, although sometimes every five days. Temperature and precipitation were daily data, which were averaged to obtain the monthly mean.

Thereafter, we investigated seasonal changes in the snow water equivalent and runoff in the three experimental basins. In particular, we focused on the snow water equivalent and runoff in spring 2012. Generally, a smaller snow water equivalent in spring will lead to drought in the following summer through smaller runoff from spring to summer. Additionally, we investigated the seasonal changes in temperature anomalies from spring to summer. Warmer spring temperatures increase snowmelt, which results in smaller runoff in the following summer. We also analyzed the seasonal changes in precipitation from spring to summer; lower precipitation during spring–summer will cause a drought.

Finally, we calculated extreme indices of temperature and precipitation for each month [22] using the daily temperature and precipitation from meteorological stations nearest to each basin ([10], Table 2 and Figure 1). Among the 27 indices proposed by the WMO (World Meteorological Organization), we calculated the robust ones listed in Table 3. These indices can be calculated not only on an annual basis, but also on a monthly basis. In this study, we discuss the results of monthly calculations. A flowchart of the research procedure is illustrated in Figure 3.

**Table 1.** Hydrometeorological data at Tomsk and three experimental basins.

| Name | Position in Western Siberia | Longitude and Latitude | Period of Temperature and Precipitation | Period of Snow * |
|---|---|---|---|---|
| (Station) Tomsk | Middle | 56.5 N, 85.0 E | 1 January 1966– 30 June 2019 | 10 January 1996– 30 December 2017 |
| (Basin) Pravaya Khetta at Pangody | North | 65.9 N, 75.0 E | 1 January 1966– 30 June 2019 | 20 January 1985– 20 December 1990 |
| Dubches at Sandakches | Middle | 61.7 N, 87.9 E | 1 January 1966– 30 June 2019 | 20 January 1985– 10 May 2019 |
| Ursul at Onguday | South | 50.8 N, 85.7 E | 1 January 1966– 30 June 2019 | 31 October 1986– 30 December 2017 |

**Table 1.** *Cont.*

| Name of Basin | Drainage Area (km$^2$) | Period of Runoff | Notes on Runoff |
|---|---|---|---|
| Pravaya Khetta at Pangody | 1.200 | 1 January 1981–1 January 1991 | Missing from 1 May 1982 to 24 May 1982 |
| Dubches at Sandakches | 8.360 | 1 October 2008–30 September 2020 | |
| Ursul at Onguday | 3.080 | 1 October 2008–30 September 2020 | |

Note: * Snow depth at Tomsk and snow water equivalent at three experimental basins [10].

**Table 2.** Meteorological data used for calculating extreme indices [10].

| Name of the Station | Position in Western Siberia | Latitude and Longitude | Period of Temperature and Precipitation |
|---|---|---|---|
| Nadym | North | 65.5 N, 72.7 E | 1 January 1961–30 June 2019 |
| Tarko-Sale | North | 64.9 N, 77.8 E | 1 January 1961–30 June 2019 |
| Bor | Middle | 61.6 N, 90.0 E | 1 January 1961–30 June 2019 |
| Ust-Koksa | South | 50.3 N, 85.6 E | 1 January 1961–30 June 2019 |

**Table 3.** Calculated extreme indices [22].

| Index | Definition | Unit |
|---|---|---|
| TXx | Maximum of daily maximum temperature (TX) | °C |
| TXn | Minimum of daily TX | °C |
| TNx | Maximum of daily minimum temperature (TN) | °C |
| TNn | Minimum of daily TN | °C |
| TX90P | Percentage of days when TX > 90th percentile of TX in 1961–1990 | % |
| TX10P | Percentage of days when TX > 10th percentile of TX in 1961–1990 | % |
| TN90P | Percentage of days when TN > 90th percentile of TN in 1961–1990 | % |
| TN10P | Percentage of days when TN > 10th percentile of TN in 1961–1990 | % |
| RX5day | Maximum 5-day precipitation | mm |
| RX1day | Maximum daily precipitation | mm |
| DTR | Diurnal temperature range | °C |

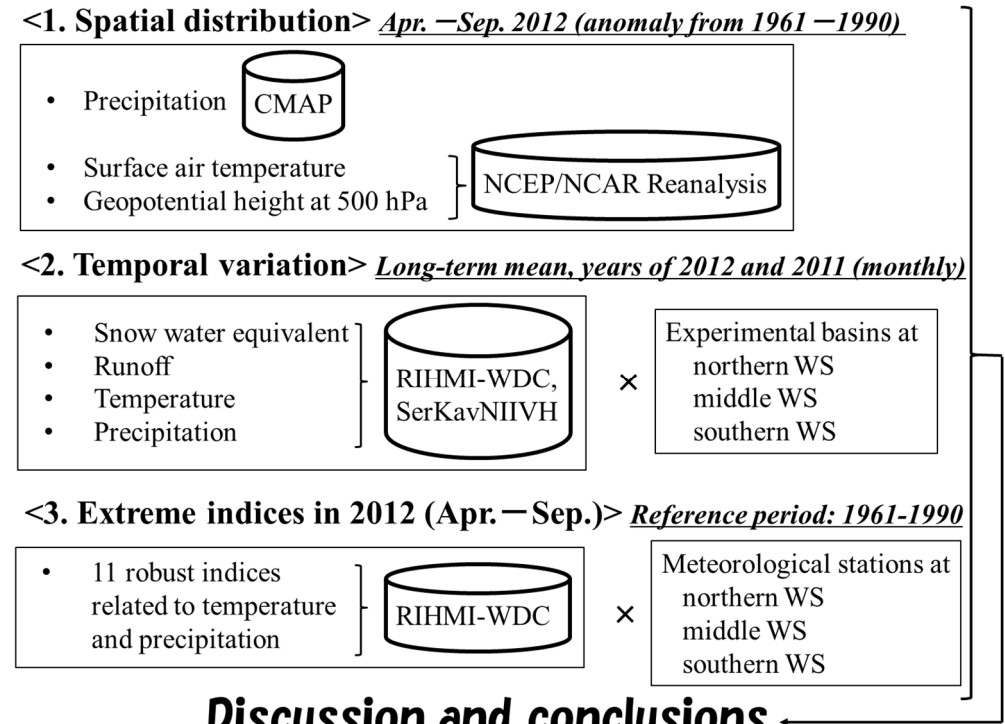

**Figure 3.** A flowchart of the research procedure of the present study. For the acronyms, refer to Section 2 and references.

### 3. Results and Discussion

*3.1. Spatial/Temporal Variation in the Atmospheric Field during Spring–Summer 2012*

Figure 4 displays the spatial/temporal variation in precipitation anomalies during 1961–1990 around WS from April to September. In April and May, precipitation anomalies around middle WS (Tomsk and Dubches at Sandakches) and south WS (Ursul at Onguday) were subtle; that is, both positive and negative anomalies co-existed. In contrast, north WS (Pravaya Khetta at Pangody) experienced a positive anomaly (Figure 3a,b). In June, the situation drastically changed. Middle WS experienced less precipitation, less than 2.0–2.5 mm/day, in comparison with the long-term mean (1961–1990). However, precipitation was not equally distributed zonally, and north and south WS received more precipitation, especially in July (Figure 4d). The harsh conditions in middle WS continued until August; it was mitigated in September by a prevailing positive anomaly (Figure 4f). Throughout the period, north WS received a positive anomaly, whereas south WS experienced both positive and negative anomalies, both being subtle.

Figure 5 depicts the temperature anomalies from April to September 2012. Except for August, positive anomalies prevailed throughout WS; that is, the study area experienced an overall high-temperature anomaly from spring to summer in 2012. In particular, middle WS around Tomsk was dominated by a positive anomaly throughout the period. The positive anomaly was especially apparent in June and its center was located in north WS (Figure 5c). This situation in June and July 2012 was also reported in [11]. Although a negative temperature anomaly prevailed from north to middle WS in August, the situation reversed in September (Figure 5e,f). When combined with Figure 4, it becomes clear that in June and July 2012, middle WS experienced hot and dry weather.

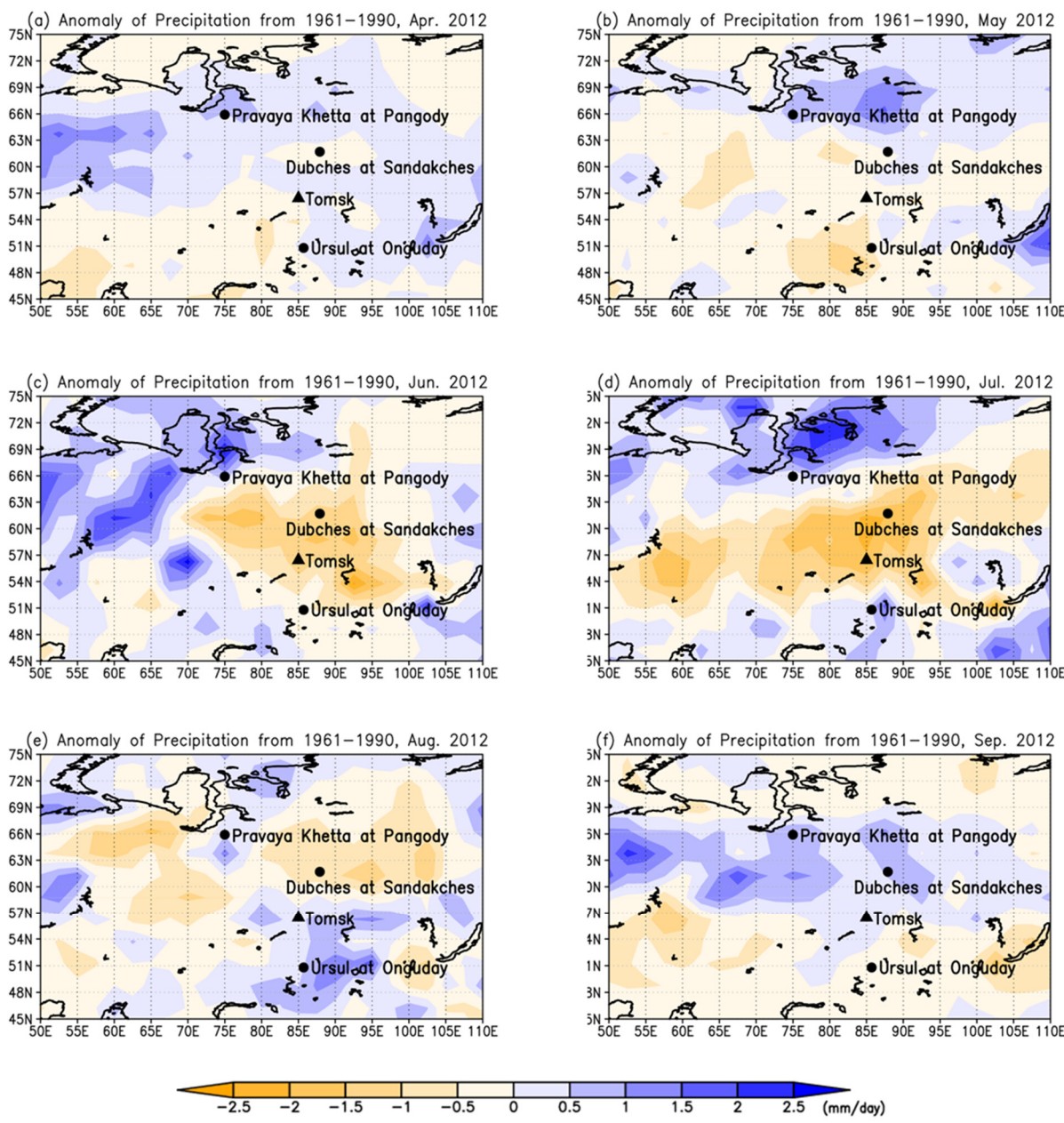

**Figure 4.** Distribution of precipitation anomaly from 1961 to 1990 in 2012, based on CMAP [18]. (**a**) April, (**b**) May, (**c**) June, (**d**) July, (**e**) August, and (**f**) September. Black circles are experimental basins, whereas black triangle represents Tomsk.

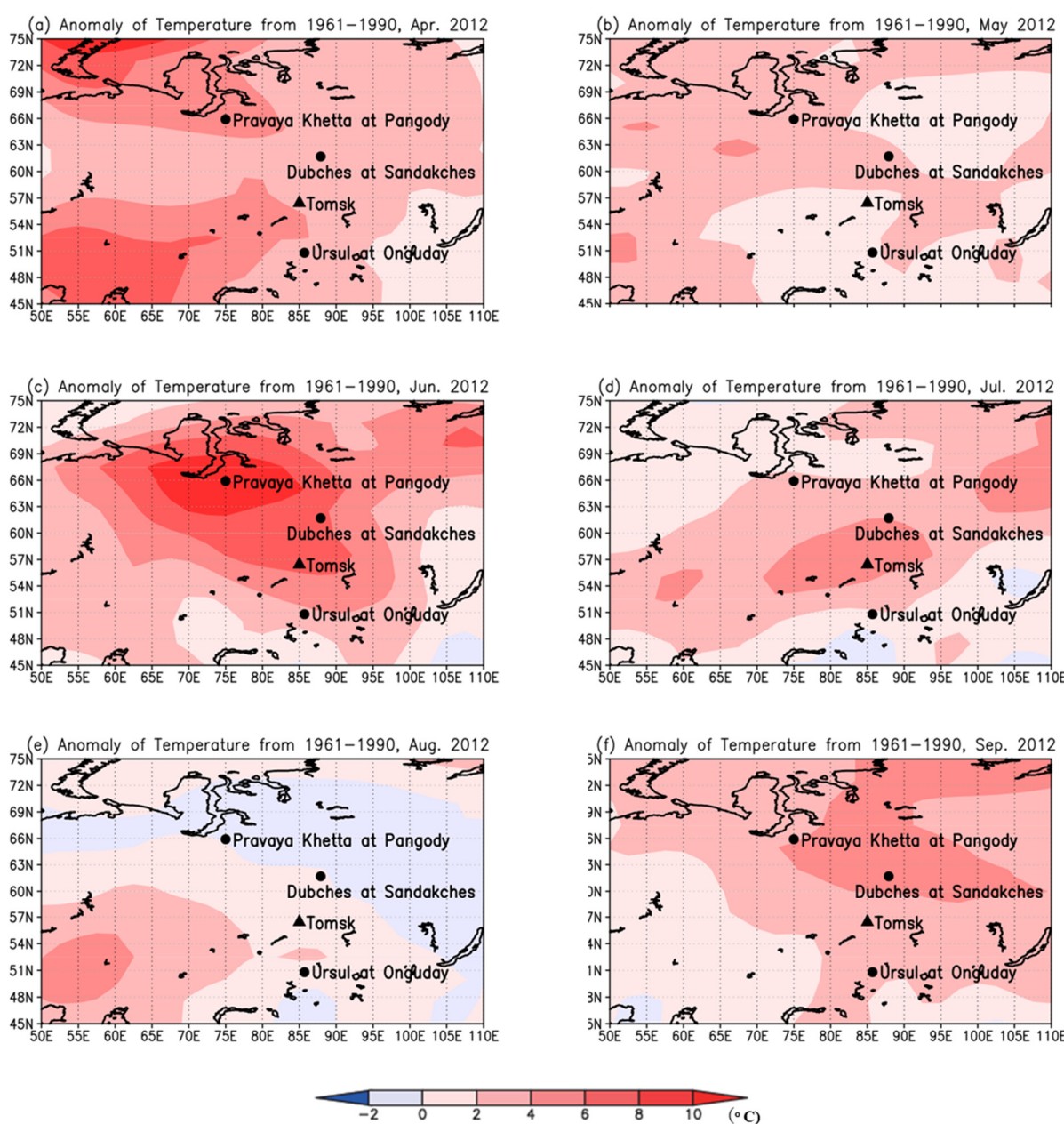

**Figure 5.** Same as Figure 4, but for surface air temperature. The data are derived from the NCEP/NCAR reanalysis [19]. Black circles are experimental basins, whereas black triangle represents Tomsk.

Figure 6 shows the geopotential height anomaly at 500 hPa from April to September 2012. The pattern resembles that in Figure 5; that is, a positive geopotential height anomaly corresponds to a positive temperature anomaly. The barotropic situation determines the temperature anomaly. The positive geopotential height anomaly also resembled the precipitation anomaly (Figure 4). Although the center of the geopotential height anomaly was located in north WS in June, north WS received substantial precipitation. We can grasp such spatial differences; however, less precipitation around middle WS corresponds well with the geopotential anomaly.

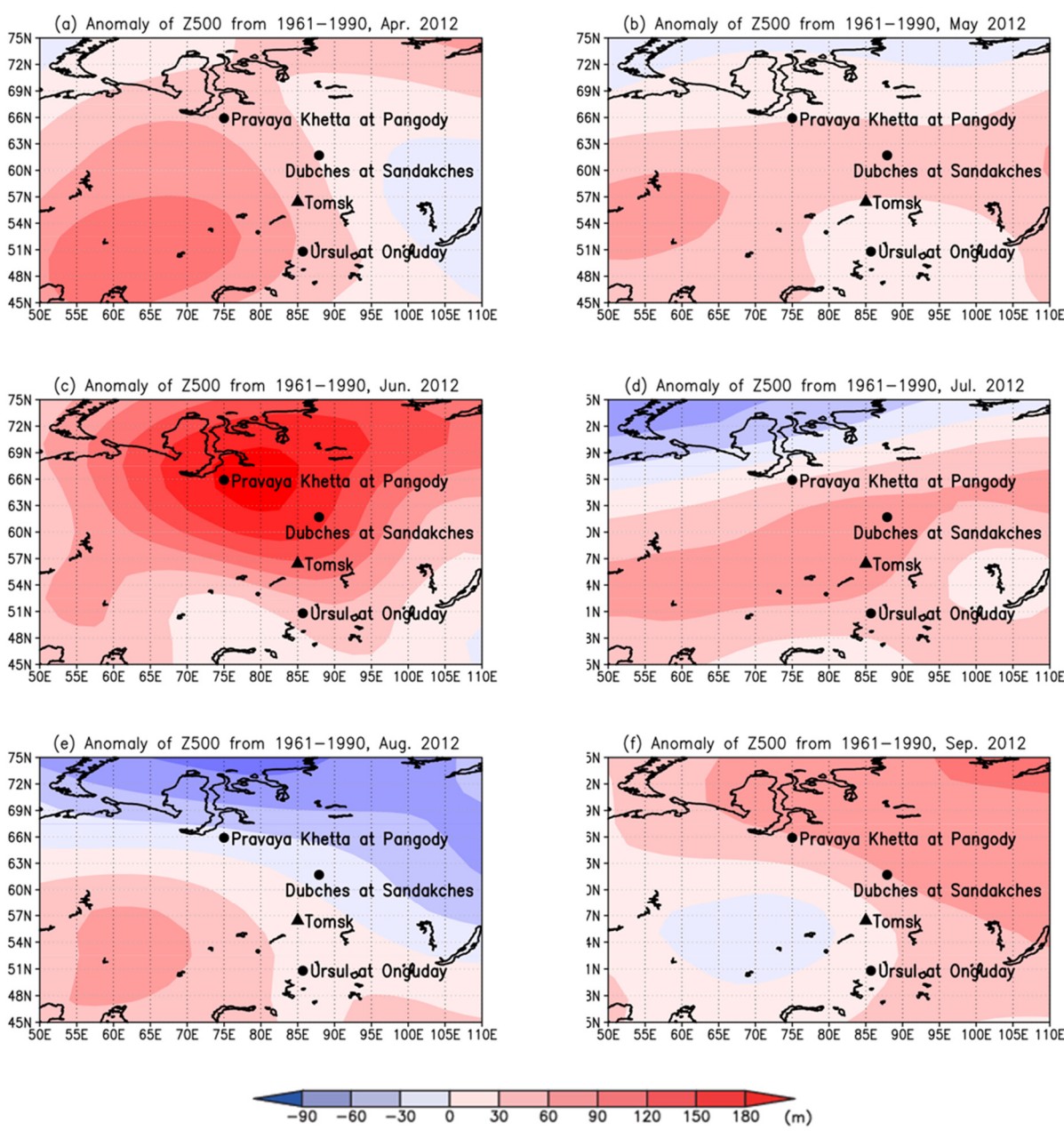

**Figure 6.** Same as Figure 4, but for geopotential height at 500 hPa. The data source is the same as for Figure 5 [19]. Black circles are experimental basins, whereas black triangle represents Tomsk.In summary, the entire WS experienced a high temperature anomaly from spring to summer in 2012; however, low precipitation was limited to middle WS, especially in June and July. In the next section, we explore the temporal variation in the hydrometeorological characteristics in middle, north, and south WS.

### 3.2. Hydrometeorological Characteristics at Dubches at Sandakches, Middle WS in 2012 and 2011

Figure 7a displays the seasonal changes in the snow water equivalent at Dubches at Sandakches, middle WS. The snow water equivalent in March 2012 was the third lowest from 1985 to 2019. The negative deviation is beyond one standard deviation; that is, the deficit of snow as a water resource was apparent in spring 2012. However, the more surprising fact is that the snow had already disappeared in May 2011 although it was still present in May 2012, the severe drought year (Figure 7a). The snow deficit observed in spring 2011 did not lead to a drought in the following summer; we discuss this at the end of this section.

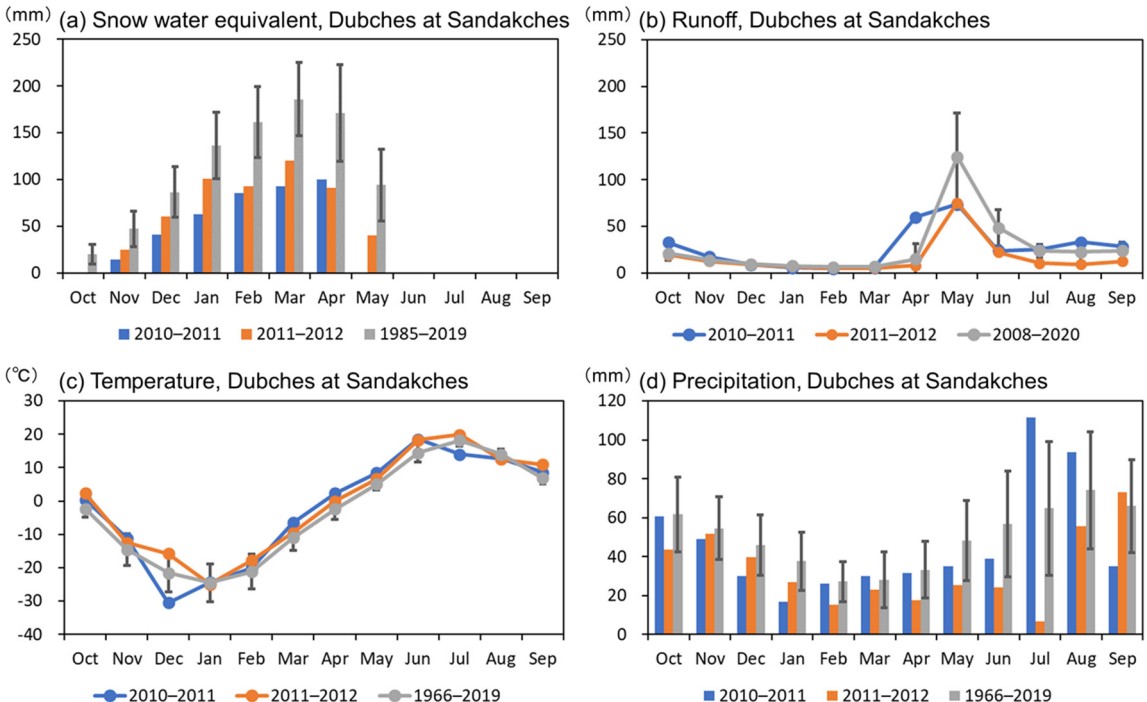

**Figure 7.** Seasonal change in hydrometeorological elements at Dubches at Sandakches in middle WS from October to September. The bars indicate the standard deviation of the long-term mean for the respective gray legends. The data for 2010–2011 and 2011–2012 are also depicted. (**a**) Snow water equivalent, (**b**) runoff, (**c**) temperature, and (**d**) precipitation.

Figure 7b illustrates the seasonal change in runoff at Dubches at Sandakches. Reflecting the snowmelt (Figure 7a), the peak runoff appears in May, although the standard deviation is the largest for the year. This implies that the runoff in May shows large inter-annual variability. For example, a large runoff was observed in April 2011 when snowmelt occurred earlier than usual (Figure 7a). This resulted in a smaller runoff in May 2011, which was almost the lower limit of one standard deviation in May (Figure 7b). The lower runoff in May 2011 was almost the same as that in May 2012. In 2012, runoff from April to September was systematically smaller than the long-term mean for 2008–2020, which represents a hydrological drought.

Figure 7c depicts the seasonal change in the monthly mean temperature at Dubches at Sandakches. As shown in the figure, the temperature during March–July 2012 was 1.5–3.9 °C higher than the long-term mean from 1966 to 2019. Although the standard deviation was small during spring–summer, the temperature during March–July 2012 exceeded the upper limit of one standard deviation (Figure 7c). The higher temperature in spring increased the snowmelt, despite the smaller snow water equivalent in spring 2012 (Figure 7a). This is one of the reasons for the severe drought in 2012.

Figure 7d shows the seasonal change in monthly precipitation at Dubches at Sandakches. Precipitation was higher in the warm season and lower in the cold season. In comparison with the long-term mean from 1966 to 2019, precipitation during April–August 2012 was 16–58 mm lower. Precipitation from April to July 2012 was below the lower limit of one standard deviation. In particular, a negative anomaly was apparent in July (Figures 4d and 7d). Although severe drought continued until August, precipitation was found to have a positive anomaly in September 2012 (Figures 4f and 7d). Lower precipitation from spring to summer, along with less snow water equivalent at the end of spring and a smaller runoff in the succeeding months, certainly led to a severe drought in the summer of 2012.

A peculiar characteristic of Figure 7 is that the smallest snow water equivalent in March 2011 during the period 1985–2019 (Figure 7a) did not lead to drought in the following summer. In 2011, the monthly mean temperature was higher than the long-term mean, such as in 2012 (Figure 7c); however, precipitation from February to August was systematically higher in 2011 than in 2012. In particular, precipitation in July and August 2011 was higher than the long-term mean (Figure 7d). It is thought that these atmospheric conditions prevented the occurrence of a drought in the summer of 2011, despite the earlier disappearance of snow at Dubches at Sandakches.

### 3.3. Hydrometeorological Characteristics at Pravaya Khetta at Pangody, North WS in 2012 and 2011

As the data for snow water equivalent from 1985 to 1990 at Pravaya Khetta at Pangody, north WS (Table 1) are limited, we cannot discuss its characteristics in 2012 and 2011 (Figure 8a). However, the seasonal change in snow water equivalent was similar to that at Dubches at Sandakches (Figure 7a). A comparison of Figures 7a and 8a reveals that snowmelt began earlier at Dubches at Sandakches. Additionally, the maximum snow water equivalent was larger at Pravaya Khetta at Pangody than at Dubches at Sandakches. These characteristics reflect the positions of the experimental basins; that is, Pravaya Khetta at Pangody is located further north than Dubches at Sandakches (Figure 1).

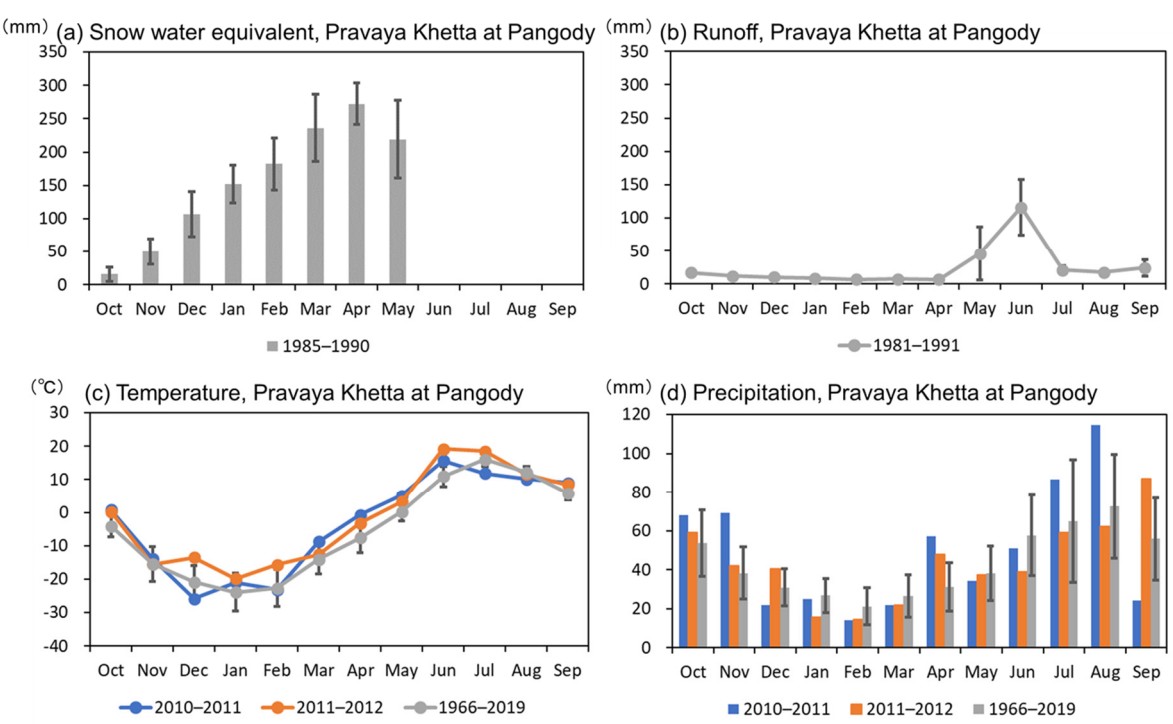

**Figure 8.** Same as Figure 7, but for Pravaya Khetta at Pangody in north WS. In (**a**) and (**b**), data for 2010–2011 and 2011–2012 are missing.

Figure 8b displays the seasonal change in runoff at Pravaya Khetta at Pangody. Reflecting snowmelt, the maximum runoff was observed in June, although the standard deviations in May and June were larger than those of the other months. This implies that the peak runoff has significant interannual variability in terms of magnitude and appearance. Owing to the limitation of data availability (Table 1), we cannot discuss the hydrological characteristics in 2012 and 2011.

Figure 8c depicts the seasonal change in temperature at Pravaya Khetta in Pangody. The figure shows that temperatures from March to July 2012 were 1.4–8.3 °C higher than the long-term mean, which was larger than the upper limit of one standard deviation. Precipitation in April and September 2012 was higher than the long-term mean (Figure 8d),

which also exceeded the upper limit of one standard deviation. In contrast, precipitation from May to August 2012 was 1–19 mm lower than the long-term mean, although it was within the range of one standard deviation. In particular, the negative anomalies in June and July were not as significant as those at Dubches at Sandakches, middle WS (Figures 7d and 8d). Therefore, it is thought that drought was not extremely apparent in north WS.

We could not use the data of the snow water equivalent and runoff at Pravaya Khetta at Pangody in 2011 (Figure 8a,b). However, precipitation in July and August 2011 was higher than the long-term mean (Figure 8d). Precipitation in August 2011 exceeded the upper limit of one standard deviation, i.e., severe drought did not occur in 2011 in north WS.

### 3.4. Hydrometeorological Characteristics at Ursul at Onguday, South WS in 2012 and 2011

Figure 9a displays the seasonal changes in the snow water equivalent at Ursul at Onguday, south WS from 1986 to 2017. The figure indicates that the maximum snow water equivalent appears in February, although a significant standard deviation is observed every month. In both 2012 and 2011, smaller snow water equivalents were observed in March, and snow disappeared in April. The snow water equivalent in March 2012 was the fifth lowest from 1986 to 2017. A more surprising fact is that the value was smaller than the lower limit of one standard deviation in April 2011; that is, the deficit of snow as a water resource was apparent in both the spring of 2012 and 2011. However, the latter occurrence did not lead to a drought in the summer of 2011; we discuss this at the end of this section.

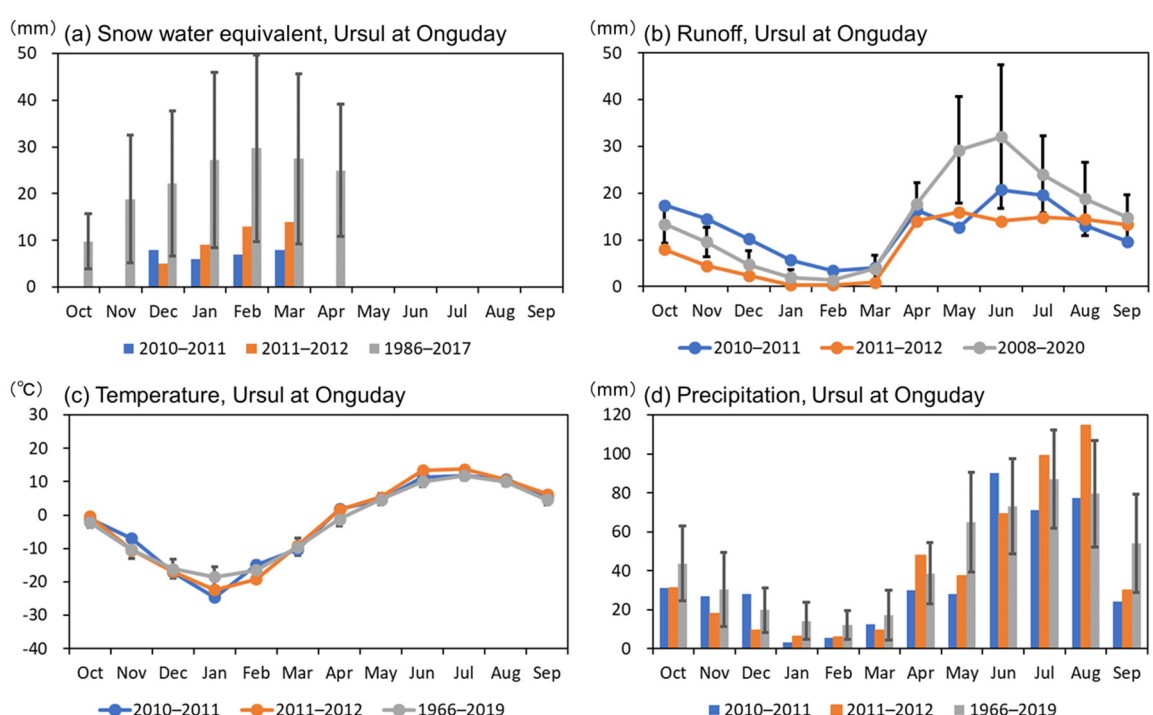

**Figure 9.** Same as Figure 7, but for Ursul at Onguday in South WS.

Figure 9b illustrates the seasonal changes in runoff at Ursul at Onguday. The maximum runoff occurred in June, although the interannual variability was also the largest in the year. Reflecting the smaller snow water equivalent in 2011 and 2012, the runoff in April to September in these years was systematically smaller than the long-term mean (Figure 9b). In May, the runoff in 2011 was smaller than that in 2012, both beyond the lower limit of one standard deviation. This reflects the characteristics of precipitation in these years; that is, precipitation in May 2011 and May 2012 were both beyond the lower limit of one standard deviation (Figure 9d). Runoff in 2011 and 2012 after July was within the range of one

standard deviation, which reflects the characteristics of precipitation during these months (Figure 9d).

Figure 9c depicts the seasonal changes in monthly mean temperature at Ursul at Onguday. The figure shows that the temperature during March–June 2012 was 0.4–3.3 °C higher than the long-term mean from 1966 to 2019. However, the temperature anomaly was not as significant as that at Dubches at Sandakches, middle WS (Figure 7c). In general, the higher temperature in spring increased snowmelt even though the snow water equivalent in March 2012 was small.

Figure 9d shows the seasonal changes in monthly precipitation at Ursul at Onguday. In comparison with the long-term mean from 1966 to 2019, precipitation in May 2012 was beyond the lower limit of one standard deviation. The anomaly changed from negative to positive from June to July 2012, and August 2012 also exhibited a positive anomaly that exceeded the upper limit of one standard deviation. This situation is different from that of Dubches at Sandakches (Figure 7d).

The snow water equivalent in March 2011 was the lowest recorded from 1986 to 2017 (Figure 9a). Although the temperature from spring to summer was higher than the long-term mean (Figure 9c), the precipitation after June 2011 was larger than or within one standard deviation of the long-term mean (Figure 9d). The smallest snow water equivalent in spring 2011 therefore did not lead to a severe drought in the summer of 2011 at Ursul at Onguday.

### 3.5. Characteritics of Extreme Indices

Table 4 lists the monthly extreme indices from April to September 2012 regarding the stations in Table 2 in descending order. The comparison was made from 1966 to 2019 based on the climate normal from 1961 to 1990. In this table, the top-five ranked indices showing dry/warm phenomena are indicated by bold and underlined, whereas those showing wet/cold ones are indicated via italics and underlined. The number of five was selected arbitrarily. Some data are missing in a few stations, for example, the rank 49 of DTR (daily temperature range) at Tarko-Sale in April 2012 is shown by italics and underlined, whereas that at Bor in April 2012 is not.

**Table 4.** Descending rank of monthly extreme indices from April to September in 2012 compared with those from 1966 to 2019. The climate normal was calculated from 1961 to 1990.

| | April 2012 | | | | May 2012 | | | | June 2012 | | | |
| | North | | Middle | South | North | | Middle | South | North | | Middle | South |
| Index | Nadym | Tarko-Sale | Bor | Ust-Koksa | Nadym | Tarko-Sale | Bor | Ust-Koksa | Nadym | Tarko-Sale | Bor | Ust-Koksa |
|---|---|---|---|---|---|---|---|---|---|---|---|---|
| TXx | 27 | 18 | 35 | 8 | **2** | **2** | 7 | 11 | **1** | **1** | 17 | **1** |
| TXn | 7 | 8 | 12 | **1** | 24 | 26 | 9 | 12 | **1** | **1** | **2** | **3** |
| TNx | 30 | 22 | 12 | 15 | **1** | **1** | 6 | *51* | **3** | **2** | 15 | **1** |
| TNn | **4** | **5** | 6 | 7 | 11 | 10 | **2** | 44 | 8 | **5** | 7 | **4** |
| TX90P | 19 | 11 | 14 | **3** | 7 | 8 | 12 | 15 | **1** | **1** | **3** | **1** |
| TX10P | 28 | 33 | 24 | 45 | 32 | 35 | 31 | 40 | 33 | 32 | 41 | 31 |
| TN90P | 8 | 7 | 19 | **4** | 6 | 14 | 7 | 23 | **1** | **2** | 10 | **1** |
| TN10P | 34 | 26 | 36 | 27 | 26 | 29 | 45 | 28 | 29 | 23 | 38 | 44 |
| RX5day | 7 | 8 | 45 | 24 | 48 | 18 | 44 | 45 | 15 | **51** | 29 | 21 |
| RX1day | 5 | 10 | 43 | 36 | 45 | 14 | **53** | **50** | 7 | 33 | 28 | 30 |
| DTR | 41 | *49* | 49 | **5** | 21 | 27 | 34 | 7 | **1** | **1** | 6 | 14 |

**Table 4.** *Cont.*

| | July 2012 | | | | August 2012 | | | | September 2012 | | | |
|---|---|---|---|---|---|---|---|---|---|---|---|---|
| | North | | Middle | South | North | | Middle | South | North | | Middle | South |
| Index | Nadym | Tarko-Sale | Bor | Ust-Koksa | Nadym | Tarko-Sale | Bor | Ust-Koksa | Nadym | Tarko-Sale | Bor | Ust-Koksa |
| TXx | 9 | 10 | 9 | **4** | 21 | *48* | 33 | 20 | 29 | 12 | 6 | 9 |
| TXn | 6 | 11 | 9 | 16 | 33 | 16 | 15 | 32 | 7 | 8 | **2** | 14 |
| TNx | 13 | **1** | 31 | 18 | 44 | 38 | 24 | 28 | 24 | **5** | **3** | 6 |
| TNn | 7 | 6 | 15 | **1** | 31 | 35 | 43 | 38 | 7 | 19 | **1** | 14 |
| TX90P | **5** | **4** | **1** | **2** | 16 | 40 | 23 | 8 | 18 | 13 | 7 | 6 |
| TX10P | 25 | 35 | 32 | 42 | 23 | 33 | 13 | 15 | 35 | 34 | 42 | *49* |
| TN90P | **4** | **4** | 19 | **3** | 44 | 38 | 39 | 17 | 11 | 8 | **1** | 25 |
| TN10P | 31 | 42 | 30 | 43 | 11 | 8 | **3** | 19 | 41 | 41 | 47 | *50* |
| RX5day | 40 | 19 | **53** | **51** | 38 | 35 | 37 | 11 | *2* | 17 | 26 | 20 |
| RX1day | 40 | 36 | **51** | 43 | 23 | 30 | 32 | 33 | 12 | 24 | *5* | 10 |
| DTR | 16 | 27 | 7 | 20 | 6 | 18 | 11 | 34 | 11 | 16 | 31 | 12 |

Note: Indices in the top-five ranks showing dry/warm phenomena are indicated in bold and underlined, whereas those showing wet/cold phenomena are indicated in italics and underlined.

The table shows that the characteristics of Bor, located in central area of WS, within the Yenisei River valley, are somewhat different from those of the other two regions. In April 2012, there were no indices within the top-five ranks at Bor; however, both RX5day (45th) and RX1day (43rd) were classified in the lower ranks; that is, precipitation in April 2012 was lower at Bor. At Nadym and Tarko-Sale, located in north WS, TNn are fourth and fifth, respectively. At Tarko-Sale, the DTR was within the last five; that is, the temperature in the north, especially TNn, was high in April 2012. At Ust-Koksa in south WS (in Altai mountains), TXn, TX90P, TN90P, and DTR were among the top-five. The temperature was also high in the south, although DTR showed opposite trends in north and south WS. As RX5day and RX1day were not classified in the lower ranks in north and south WS, the possibility of drought was not apparent in April in these regions.

In May 2012, TNn was high, along with small RX5day and RX1day values at Bor. At Nadym and Tarko-Sale, however, precipitation-related indices from April to May were not as distinct, such as those at at Bor. In contrast, RX1day at Ust-Koksa was within the last five ranks, as it was at Bor (Table 4). In June 2012, TXn and TX90P were high in Bor, whereas many temperature-related indices were within the top-five ranks in the other two regions. Additionally, RX5day at Tarko-Sale was distinct. In July 2012 at Bor, TX90P was high, along with smaller RX5day and RX1day values. The situation was the same as that for RX5day in Ust-Koksa. In August 2012, TN10P was high at Bor, along with high TXn, TNx, TNn, and TN90P in September 2012; however, RX1day was also high in September 2012, which terminated the severe drought in the summer. At Nadym, RX5day was large; however, the temperature was not as high as in September 2012 compared to Bor. Additionally, in Ust-Koksa, the temperature in September 2012 was not too high.

In summary, even though global warming trends were considered, the entire WS experienced a high-temperature anomaly from spring to summer in 2012, especially in June and July. However, north and south WS did not suffer from severe drought in 2012 because substantial precipitation was observed in summer. The extreme precipitation indices in 2012 in north and south WS were not as distinct as those in middle WS. In middle WS, five-day precipitation and daily precipitation in July 2012 were the lowest recorded from 1966 to 2019. These characteristics were unique to middle WS, and resulted in a severe drought in the region.

## 4. Conclusions

Tomsk, located in middle WS, suffered from a severe drought in summer 2012. The objective of this study was to clarify the regional differences in hydrometeorological parameters in WS in 2012. At first, we investigated the spatial/temporal variation in the precipitation, surface air temperature, and geopotential height at 500 hPa in April–September 2012 to reconfirm the severe drought in middle WS. We then investigated the yearly hydrometeorological parameters using snow water equivalent, temperature, and precipitation data from northern, middle and southern experimental basins in WS. We also calculated some extreme indices at four meteorological stations on a monthly basis. The findings of this study are summarized as follows:

1. The entire WS experienced a high temperature anomaly from spring to summer in 2012, especially in June and July.
2. In middle WS, the snow water equivalent in March 2012 was the third smallest from 1985 to 2019. The runoff in spring was also systematically lower than the long-term mean. Precipitation during April–August 2012 was continuously lower than the long-term mean. All these factors resulted in a severe drought. In particular, the five-day precipitation and daily precipitation in July 2012 were the lowest from 1966 to 2019. These characteristics were unique to middle WS alone, including Tomsk.
3. In middle WS in 2011, the snow water equivalent in March was the lowest from 1985 to 2019. The peak runoff also appeared earlier than normal in 2011. However, precipitation during July–August 2011 prevented the occurrence of a severe drought in the summer of 2011.
4. North and south WS did not suffer from severe drought in 2012 because substantial precipitation was observed in the summer, although the snow water equivalent in March 2012 in south WS was the fifth lowest from 1986 to 2017. The extreme precipitation indices in 2012 in north and south WS were not as distinct as those in middle WS.

Finally, we should reconsider the characteristics of the severe drought around Tomsk in 2012 (Figure 2). The conclusion is that the severe drought was limited to and unique to middle WS in comparison with the other regions in WS. The findings of this study will certainly contribute to the prediction of future hydrometeorological events, as extreme phenomena are more likely to occur in accordance with the progress of global warming.

As a remaining problem, we have to clarify why and how abnormal atmospheric conditions occurred, continued, and terminated in 2012 (Figures 4–6). The atmospheric effect on the terrestrial hydrological cycle was, to some extent, clarified by the present study; however, the temporal variation in the atmospheric conditions and its forcing mechanism has not been fully clarified. This is a remaining problem in the future.

**Author Contributions:** Conceptualization, H.M.; methodology, H.M.; software, T.W.; validation, H.M.; formal analysis, H.M.; investigation, T.W.; resources, V.Z.; data curation, T.W.; writing—original draft preparation, H.M.; writing—review and editing, T.W. and V.Z.; visualization, H.M.; supervision, H.M.; project administration, H.M.; funding acquisition, H.M. and V.Z. All authors have read and agreed to the published version of the manuscript.

**Funding:** This research was funded by JST/SICORP (JPMJSC1901, PI: H.M.) and JSPS KAKENHI (18KK002800, PI: H.M.). V.Z. was supported by a Russian Federal Targeted Program Grant (RFMEFI-61419X0002).

**Institutional Rview Board Statement:** Not applicable.

**Informed Consent Statement:** Not applicable.

**Data Availability Statement:** Temperature, precipitation, and snow depth/snow water equivalent data are available at http://aisori.meteo.ru/ClimateR (accessed on 27 April 2019) [10]. The runoff data are available at https://gmvo.skniivh.ru/ (accessed on 22 December 2022) [21].

**Conflicts of Interest:** The authors declare no conflict of interest.

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
