# Peer review of "Extreme Drought around Tomsk, Russia in Summer 2012 in Comparison with Other Regions in Western Siberia"

_water, doi:10.3390/w15030388_

Round 1

Reviewer 1 Report

The article found that the runoff during April–September 2012 was smaller than the long-term mean. Precipitation during April–August 2012 was also continuously lower. All these resulted in severe drought in summer. Particularly, precipitation in July 2012 in central WS was in the lowest class for the period 1966–2019. These characteristics were unique to central WS in July 2012.

The article is interesting; however, it required some changes before publication.

1. Please start the abstract with the main objectives of the study, and write main policy implications at the end of abstract.

2. I admire the written novelty of the manuscript. It will be great if author add few information regarding the consequences of extreme weather on agricultural damages. Author may follow these studies to highlight the consequences of extreme weather on agricultural damages in the first paragraph of introduction.

[1] Loss and damage estimation for extreme weather events: state of the practice

[2] Extreme weather events risk to crop-production and the adaptation of innovative management strategies to mitigate the risk: A retrospective survey of rural Punjab, Pakistan

[3] Crop vulnerability to weather and climate risk: Analysis of interacting systems and adaptation efficacy for sustainable crop production

4. Please change the color of text in figure 1 to make it more visible.

5. If possible, you should remove figure 2 from the introduction, and add the data of this figure in the form of text.

6. The figure 1 should move to the section of methodology.

7. Please write the main research questions and contribution at the end of introduction section.

8. Results and discussion have written very well.

9. The structure of conclusion section should improve as: Main objectives; Use methods; and then Summarize main findings of the study.   

10. Please add main limitations and recommendations for future studies at the end of conclusion section. 

Reviewer 2 Report

The paper overall looks good.  however, these remarks must be considered before any eventual publication.

1-The paper title must be changer and to be short for relcting the main edia of this paper.

2-The abstract must be precise and sumarize clearly the content of the paper.

3-The introduction must be enhanced by following those steps:

  -Present the motivation and context of the study.

  -Define the research problem and the field of application.

  -Add relevant case study with important results.

  -Add  short sentences that introduce your results.

4-A literature review section (related works) is strongly recommended to position the contribution in relation to the existing

5-Improved English correction.

Reviewer 3 Report

Dear Editor, Dear Authors,     

The paper entitled "Extreme Drought around Tomsk, Russia in Summer 2012 in Comparison with Other Regions in Western Siberia" addresses very important issues related to climate change.  

The research problem has been very interestingly rearranged and described, however, it needs to be sorted out technically. 

After reading the article, I have the following comments and suggestions for improving the article:   

Structure of the article.   

I suggest improving the structure of the article in accordance with the journal's guidelines.   

A new numbering of chapters should be introduced   

1. Introduction   

2. Literature review/theoretical background   

3. Materials and methods   

4. Results   

5. Discussion  

6. Conclusion

Abstract   

I propose to improve to make it more readable. There is no information about the research methods used. There is also a lack of a clearly defined research objective. 

In the Introduction   

In my opinion, it should be expanded to include the following news: why was this study undertaken? What research has been done so far, where? What conclusions have been drawn from these studies. Is this article a continuation of previous research?   

At the end of the chapter should be the purpose of the research and the research questions.  

I suggest moving Figure 1 study area and Figure 2 to the Date and Methods subsection. This subchapter should end with a research question . Also missing is a well-formulated research objective. 

The article lacks subsection 2. Literature review

The article is based on only 19 items of literature, I think this is very little!

In "Materials and methods   

The authors present the research methods too generally, without a detailed description of what the research will consist of?  A diagram of the research procedure is missing.  

In Resultats  

The results are presented and described in a good way, they are very interesting and important for further research on climate change. 

In Discussion     

This section still needs to answer the question: what tangible benefits did this study bring? The authors should compare their project and results with the results of similar ongoing studies on this topic from other parts of Europe and around the world. 

Technical errors to be removed: 

Correct the literature according to the rules of the journal. 

 In conclusion, I recommend this work for publication in the journal "Water " after minor changes. 

Kind regards,

Round 2

Reviewer 1 Report

The authors have addressed all previous comments and now paper is ready to be published. 

Author Response

Thank you very much for your comment.